# Impact of Infliximab Biosimilar CT-P13 Dose and Infusion Interval on Real-World Drug Survival and Effectiveness in Patients with Ankylosing Spondylitis

**DOI:** 10.3390/jcm10194568

**Published:** 2021-10-01

**Authors:** Shin-Seok Lee, Tae-Hwan Kim, Won Park, Yeong-Wook Song, Chang-Hee Suh, Soo-Kyoung Kim, Dae-Hyun Yoo

**Affiliations:** 1Department of Rheumatology, Chonnam National University Medical School and Hospital, 42 Jebong-ro, Dong-gu, Gwangju 61469, Korea; shinseok@chonnam.ac.kr; 2Department of Rheumatology, Hanyang University Hospital for Rheumatic Diseases, 222-1 Wangsimni-ro, Seongdong-gu, Seoul 04763, Korea; thkim@hanyang.ac.kr; 3School of Medicine, Inha University, Jung-Seok Building C Dong, 605 Ho. 366 Seohae-daero, Jung-gu, Incheon 22332, Korea; parkwon@inha.ac.kr; 4Department of Internal Medicine, Division of Rheumatology, Seoul National University Hospital, 101 Daehak-ro, Jongno-gu, Seoul 110-744, Korea; ysong@snu.ac.kr; 5Medical Research Center, Institute of Human-Environment Interface Biology, Seoul National University, 103 Daehak-ro, Ihwa-dong, Jongno-gu, Seoul 03080, Korea; 6Department of Rheumatology, Ajou University School of Medicine, 164 Worldcup-ro, Yeongtong-gu, Suwon 16499, Korea; chsuh@ajou.ac.kr; 7Medical Affairs Department, Celltrion Healthcare Co., Ltd., 4th Floor, 19 Academy-ro, 51 beon gil, Yeonsu-gu, Incheon 22014, Korea; sookyoung.kim@celltrionhc.com; 8 Hanyang University Institute for Rheumatology Research, 222-1 Wangsimni-ro, Seongdong-gu, Seoul 04763, Korea

**Keywords:** ankylosing spondylitis, biosimilar, CT-P13, dose adjustment, drug survival, effectiveness, infliximab, infliximab biosimilar, infusion interval adjustment, real-world data

## Abstract

CT-P13 is an infliximab biosimilar approved for indications including ankylosing spondylitis (AS); the approved maintenance regimen is 5 mg/kg infused every 6–8 weeks. In clinical practice, modifications to infliximab dose and/or infusion interval can be beneficial to the patient. For CT-P13, real-world data on dose and/or interval adjustment are lacking. This analysis investigated the impact of such treatment pattern changes on effectiveness and drug survival up to five years for adult (≥18 years old) patients with AS in the Korean, real-world, retrospective rheumatoid arthritis and ankylosing spondylitis (RAAS) study. Overall, 337 patients with AS were identified: 219 who initiated infliximab treatment with CT-P13 (‘naïve’) and 118 who switched from reference infliximab to CT-P13 (‘switched’). Overall, 18/235 (7.7%), 110/224 (49.1%), and 101/186 (54.3%) evaluable patients had dose, infusion interval, or combined treatment pattern changes, respectively. More naïve (61.0%) versus switched (42.6%) patients had treatment pattern changes. Overall, Bath Ankylosing Spondylitis Disease Activity Index scores decreased from baseline to week 54, then remained stable; improvements were greater for patients with than without treatment pattern changes. Drug survival did not differ significantly between patients with or without treatment pattern changes. Findings suggest that adjusting dose and/or infusion interval can improve clinical outcomes for CT-P13-treated patients with AS.

## 1. Introduction

The infliximab biosimilar CT-P13 has received regulatory approval in 98 countries as of March 2021. In patients with ankylosing spondylitis (AS), the Program evaLuating the Autoimmune disease iNvEstigational drug cT-p13 in AS patients (PLANETAS) study demonstrated pharmacokinetic equivalence between reference infliximab and CT-P13, both administered using a maintenance regimen of 5 mg/kg once every eight weeks [1]. Comparability of efficacy and safety profiles was also demonstrated [1]. In the one-year open-label extension, switching from reference infliximab to CT-P13 was shown to be safe, with no adverse impact on efficacy [2]. The intravenous (IV) formulation of CT-P13 is licensed for use in the same indications as reference infliximab by the United States (U.S.) Food and Drug Administration, European Medicines Agency (EMA), and the Ministry of Food and Drug Safety in the Republic of Korea [3,4,5,6,7,8]. For AS, the approved maintenance regimen is 5 mg/kg infused once every six weeks [3] or once every 6–8 weeks [4,6].

Studies show that initiating lower-dose (3 mg/kg) infliximab treatment can be effective for the treatment of AS [9,10,11,12,13,14,15]. Positive outcomes with infliximab dose modification have also been observed in other rheumatic diseases. In patients with psoriatic arthritis, initiating infliximab treatment at doses lower than 5 mg/kg did not significantly impact drug survival or treatment response [16,17], while in patients with rheumatoid arthritis (RA) initiating lower-dose infliximab treatment, limited infliximab dose escalations (by increasing dose or decreasing infusion interval) have been associated with improved clinical responses and drug survival [18,19]. In patients with AS, individualized dose/interval adjustments based on treatment response may be warranted to achieve clinical targets in some cases [11,12,13,15]. In addition, lower-dose infliximab has been demonstrated to be well-tolerated in patients with AS [9,11,12,13,14,15], and is associated with significantly lower health care system costs [12,15]. To our knowledge, the potential benefits of dose and/or interval adjustments have not been investigated for an infliximab biosimilar treatment in AS to date. In addition, real-world data are not currently available regarding dose and/or interval adjustments in patients with AS who have switched from reference to biosimilar infliximab.

The rheumatoid arthritis and ankylosing spondylitis (RAAS) study was a real-world, retrospective analysis aiming to provide long-term (up to 5-year) data on CT-P13 drug survival, safety, and efficacy in patients with RA and AS [20]. The current analysis investigated the impact of changes to CT-P13 treatment patterns (doses and infusion intervals) on effectiveness and drug survival for patients with AS in the RAAS study. This included patients who were infliximab-naïve at CT-P13 initiation or those who had a non-medical switch from reference infliximab to CT-P13.

## 2. Materials and Methods

### 2.1. RAAS Study Design and Patients

As reported previously [20], the RAAS study was a non-interventional, retrospective, multicenter analysis of medical records for adult (≥18 years old) patients with RA or AS conducted at five referral university hospitals in the Republic of Korea. This ad hoc analysis included patients with AS who had received ≥1 dose of CT-P13 between 1 September 2012 and 31 December 2017, and were either reference infliximab-naïve at CT-P13 initiation (naïve patients) or had a non-medical switch to CT-P13 from reference infliximab (switched patients).

### 2.2. Dose and Infusion Interval Analysis

Patients were analyzed by baseline dose and divided into three groups (<4 mg/kg; ≥4–<5 mg/kg; ≥5 mg/kg). The baseline dose was defined as the third infusion dose for naïve patients (induction period dosing was excluded from the analysis for comparability with switched patients) or the first infusion dose for switched patients. If the dose remained within the same grouping (<4 mg/kg; ≥4–<5 mg/kg; ≥5 mg/kg) between baseline and the last follow-up, patients were analyzed in the constant dose group. If the dose did not remain within the same grouping between baseline and the last follow-up, patients were analyzed in the changed dose group, comprising the increased dose group (last follow-up dose grouping > baseline dose grouping) and the decreased dose group (last follow-up dose grouping < baseline dose grouping). 

Baseline infusion intervals (in weeks) were calculated as the average of the three infusion intervals after the baseline dose, and the follow-up infusion interval was the average of the last three infusion intervals. If the follow-up interval was unchanged from baseline (a difference of less than one week between baseline and follow-up infusion intervals), patients were analyzed in the constant interval group. If there were changes in infusion interval, patients were analyzed in the changed interval group. This comprised the increased interval group (where the follow-up infusion interval was at least one week longer than the baseline infusion interval) and the decreased interval group (where the follow-up infusion interval was shorter than the baseline infusion interval by at least one week).

In addition, patients with both constant dose and infusion interval (combined constant group) were compared with those with changes in dose and/or infusion interval (combined changed group).

### 2.3. Assessments

As reported previously [20], demographic data were assessed at baseline immediately prior to CT-P13 dosing. Disease characteristics, including Bath Ankylosing Spondylitis Disease Activity Index (BASDAI) score, were assessed prior to CT-P13 dosing at each clinical visit [20]. Time points for BASDAI score analysis were selected considering differences in measurement schedule between patients [20]. Data on infliximab treatment history (to determine whether patients were naïve or switched) and CT-P13 dosing (weekly date of administration, body weight at each dose, and dose received) were also collected [20].

### 2.4. Statistical Analyses

Baseline demographics and disease characteristics were compared using the Kruskal–Wallis test, chi-squared test, one-way analysis of variance, Fisher’s exact test, or Wilcoxon’s rank sum test. Drug survival was assessed by Kaplan–Meier analysis; this was based on the period of time between the first and last doses of CT-P13 for patients who discontinued CT-P13 [20]. Patients who did not discontinue CT-P13, or discontinued CT-P13 because of pregnancy or remission, were censored at the last follow-up date. Drug survival was analyzed statistically by log-rank test. All statistical analyses were performed using SAS version 9.4 (SAS Institute, Cary, NC, USA).

## 3. Results

### 3.1. Patients

Overall, 337 patients with AS (219 naïve; 118 switched) were identified in the RAAS study. Among the patients with baseline dose data, overall, most patients were male (74.8%) (Table 1).

Follow-up dose and infusion interval information were available for 235 and 224 patients, respectively (Figure 1a,b). Out of 235 evaluable patients, 18 (7.7%) had dose changes (increased dose: 12; decreased dose: 6). Of the 224 evaluable patients, 110 (49.1%) had interval changes (increased interval: 79; decreased interval: 31). In the combined analysis, 85 of 186 (45.7%) evaluable patients did not have dose and/or interval changes and comprised the combined constant group (Figure 1c). A greater proportion of naïve (61.0%) versus switched (42.6%) patients had dose and/or interval changes; these patients formed the combined changed group.

### 3.2. Dose Analyses

In total, 71, 117, and 82 patients were included in the <4 mg/kg, ≥4–<5 mg/kg, and ≥5 mg/kg baseline dose groups, respectively. Most baseline demographics and disease characteristics were comparable between the baseline dose groups overall (Table 1) and for patients in the naïve (Appendix A) and switched (Appendix A) groups, although there were statistically significant differences in age between the baseline dose groups, both overall and for naïve or switched patients. Patients in the <4 mg/kg baseline dose group had the lowest median BASDAI scores overall (Table 1), although scores differed substantially between naïve patients (7.00; Appendix A) and switched patients (0.57; Appendix A). Switched patients had lower baseline BASDAI scores than naïve patients across baseline dose groups (Appendix A). Naïve patients with lower baseline doses tended to have lower baseline BASDAI scores, although differences between groups were not significant (Appendix A). For switched patients, differences in baseline BASDAI score were significantly different between baseline dose groups, with median scores of 0.57, 1.01, and 4.24 in the <4 mg/kg, ≥4–<5 mg/kg, and ≥5 mg/kg baseline dose groups, respectively (Appendix A).

Following initial decreases as naïve patients initiated CT-P13, median BASDAI scores remained consistent over time (Table 2). For the ≥4–<5 mg/kg and ≥5 mg/kg baseline dose groups, post-baseline median BASDAI scores were broadly comparable between naïve and switched patients. In the <4 mg/kg baseline dose group, naïve patients tended to have higher post-baseline median BASDAI scores than switched patients.

Mean cumulative annual maintenance doses were similar through Years 1–4 and were similar between naïve and switched patients throughout (Figure 2). Cumulative doses were lower for Year 5, reflecting the few evaluable patients. Drug survival did not differ significantly between baseline dose groups overall or for naïve patients (Figure 3a,b). For switched patients, drug survival appeared numerically lower in the ≥5 mg/kg baseline dose group compared with the other dose groups, but these differences were not statistically different (Figure 3c).

Most baseline demographics and disease characteristics were comparable between constant dose, increased dose, and decreased dose groups, overall (Appendix A) and for naïve (Appendix A) and switched (Appendix A) patients.

### 3.3. Infusion Interval Analyses

Median (IQR) baseline infusion intervals were 8.0 (7.0–9.0) weeks, 8.0 (6.0–8.0) weeks, and 8.0 (8.0–9.0) weeks in the overall (*n* = 303), naïve (*n* = 192), and switched (*n* = 111) groups, respectively. Mean infusion intervals increased slightly between Years 1 and 3 in all groups before decreasing in Year 4 (particularly in the naïve group) (Figure 2). Consistent with the baseline findings, switched patients had longer infusion intervals than naïve patients throughout. Median annual infusion numbers were comparable between groups throughout the study period and remained consistent between Years 1 and 4 (Appendix A). These were reduced in Year 5, when the number of evaluable patients was lower.

Most baseline demographics and disease characteristics were comparable between constant interval, increased interval, and decreased interval groups, overall (Appendix A) and for naïve (Appendix A) and switched (Appendix A) patients. Switched patients in the constant interval group were relatively evenly distributed between baseline dose groups. In the increased interval group, 58.3% of switched patients were in the <4 mg/kg baseline dose group, while 69.2% of switched patients with decreased infusion intervals had a baseline dose of ≥4–<5 mg/kg. Naïve patients were most commonly in the ≥4–<5 mg/kg baseline dose group, regardless of interval analysis group.

### 3.4. Combination Analyses

Most baseline characteristics were comparable between the combined constant and combined changed groups, overall (Appendix A) and for naïve (Appendix A) and switched (Appendix A) patients. However, median baseline C-reactive protein and erythrocyte sedimentation rate were significantly higher overall in the combined changed than in the combined constant group. There were no significant differences in BASDAI scores between the combined constant and combined changed groups overall and for switched patients, but baseline BASDAI scores differed significantly between these groups for naïve patients. Overall, a similar proportion of patients in the <4 mg/kg baseline dose group were in the combined constant (24.7%) and combined changed (30.7%) groups. Switched patients in the combined constant group were most likely to be in the ≥5 mg/kg baseline dose group (43.6% patients). In the combined changed group, only 13.8% of switched patients were in the ≥5 mg/kg baseline dose group. Naïve patients were most likely to be in the ≥4–<5 mg/kg baseline dose group, regardless of the combined analysis group.

After decreases from baseline to week 54 overall and for naïve patients, BASDAI scores remained stable throughout the study (Figure 4 and Appendix A). Patients in the combined changed versus the combined constant group had greater improvements in BASDAI score over time. There were no significant differences in drug survival between the combined constant and combined changed groups overall or for naïve or switched patients (Figure 5).

## 4. Discussion

This analysis provides valuable long-term data on CT-P13 treatment patterns in patients with AS in routine clinical practice. In our analysis, just over half of the evaluable patients maintained a constant CT-P13 dose and infusion interval throughout, while changes to infusion interval were more common than changes to dose. Greater improvements in BASDAI scores observed in the combined changed versus combined constant group suggest that adjusting dose and/or infusion interval can improve clinical outcomes for patients with AS receiving CT-P13 treatment. Drug survival was comparable between baseline dose groups, with no adverse impact of lower baseline doses. In addition, there was no significant effect of dose and/or interval adjustments on drug survival, as this was comparable between the combined constant and combined changed groups.

Overall, 54.3% of patients maintained their baseline dose and infusion interval throughout the period of CT-P13 treatment captured in the RAAS study (up to five years). Treatment pattern changes were more common for naïve than switched patients, reflecting the need to optimize the CT-P13 regimen in patients newly initiating infliximab therapy. Cumulative annual maintenance doses were similar for naïve and switched patients throughout, although switched patients had longer infusion intervals. Overall, similar proportions of patients in the lowest baseline dose group (<4 mg/kg) either did or did not have treatment changes during the study, reflecting the findings of previous studies, demonstrating that initiating lower-dose infliximab treatment (3 mg/kg) can be effective [9,10,11,12,13,14,15], but that treatment pattern changes may be required for some patients [11,12,13,15]. While dose escalation may be more common for infliximab than other tumor necrosis factor inhibitors (TNFis) in patients with AS, this could be explained by the ease of adjusting IV-administered doses of reference infliximab compared with the fixed doses inherent to the subcutaneous (SC) administration of other TNFis [22].

In this analysis, median BASDAI scores were reduced following CT-P13 treatment for patients overall and in naïve treatment groups, regardless of the baseline dose group, while scores were maintained over time for switched patients. This supports previous reports of the effectiveness of infliximab treatment at doses lower than the approved regimen for patients with AS [9,10,11,12,13,14,15]. BASDAI scores remained stable both in the combined constant and combined changed groups, after initial decreases for naïve patients (reflected in the overall population). This is consistent with findings from the RAAS study overall [20]. There was no initial decrease in BASDAI scores for switched patients, reflecting the possibility of most switched patients having achieved disease control with reference infliximab prior to undergoing a non-medical switch to CT-P13. Consistent BASDAI scores demonstrate that disease activity was well controlled regardless of treatment pattern changes or infliximab treatment history. This supports the lack of impact of switching from reference infliximab to CT-P13 on efficacy/effectiveness demonstrated for patients with AS [2,20,23] and in studies including patients with AS, axial spondyloarthritis, or spondyloarthritis [24,25,26,27].

Drug survival was comparable regardless of the baseline dose or treatment pattern changes. For switched patients, drug survival appeared numerically lower for patients in the ≥5 mg/kg baseline dose group; however, there were no statistically significant differences between baseline dose groups. This subset of switched patients in the ≥5 mg/kg baseline dose group may not have had as well-controlled disease as did patients in the other baseline dose groups prior to switching to CT-P13 (demonstrated by higher baseline BASDAI scores). This may account for the numerically lower drug survival. The comparability of drug survival regardless of baseline dose or treatment pattern changes complements the previous report from the RAAS study demonstrating no significant difference in CT-P13 drug survival between naïve and switched patients with AS [20]. Overall 5-year drug survival in this study exceeded 60%, broadly in keeping with previous reports for reference infliximab [28,29,30,31,32,33], and reflecting the comparability of drug retention demonstrated between CT-P13 and reference infliximab after up to four years in the Korean College of Rheumatology Biologics Registry [34]. Drug survival in this study was somewhat higher than the 50% reported after four years of low-dose (3 mg/kg) reference infliximab treatment in a Canadian prospective observational analysis [14].

The current analysis relied on medical records for data collection, which may have been incomplete, limiting the conclusions that can be drawn [20]. Since some patients did not visit the hospital at regular intervals, implementation of regular interval adjustment windows was required for the analysis of BASDAI scores. In addition, there were few evaluable patients in Year 5, although the overall population and naïve and switched groups were large. The analysis was conducted in a single country, although it included patients treated at five university hospitals. Taken together, the sample size and setting provide good representation of biologic-treated patients with AS in the Republic of Korea. Safety analyses were not conducted as part of this investigation; however, previous reports have demonstrated the safety of CT-P13 treatment for patients with AS or spondyloarthritis, including after switching from reference infliximab [2,20,24,25,35].

In conclusion, our findings suggest that an individualized treatment approach—through adjusting dose and/or infusion interval—can improve clinical outcomes for CT-P13-treated patients with AS, without adversely affecting drug survival. This provides valuable information for the management of biosimilar treatment for patients with AS in clinical practice.

## Figures and Tables

**Figure 1 jcm-10-04568-f001:**
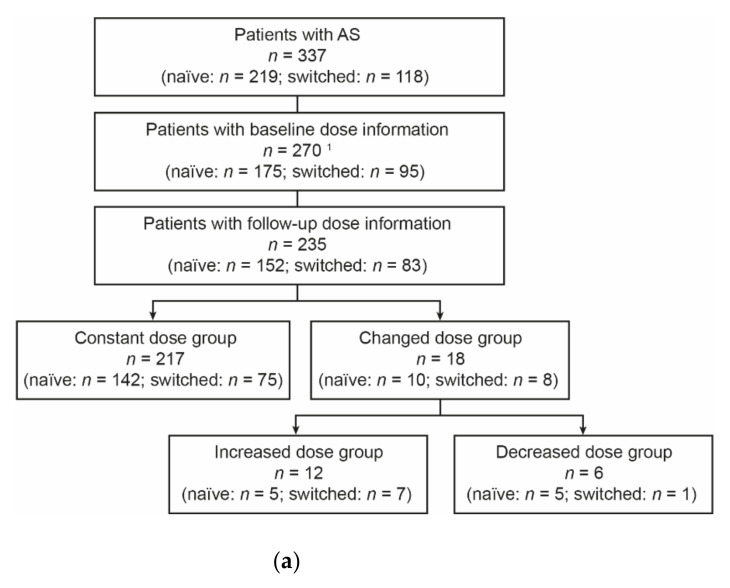
Analysis groups: (**a**) dose analysis groups, (**b**) interval analysis groups, and (**c**) combined dose/interval analysis groups. ^1^ Baseline dose was missing for 67 patients due to missing weight information (*n* = 57) or missing dose after the induction period in naïve patients (*n* = 10). ^2^ Baseline infusion interval was missing for 34 patients as there was no fourth infusion for naïve patients (*n* = 27) or no second infusion for switched patients (*n* = 7). AS, ankylosing spondylitis.

**Figure 2 jcm-10-04568-f002:**
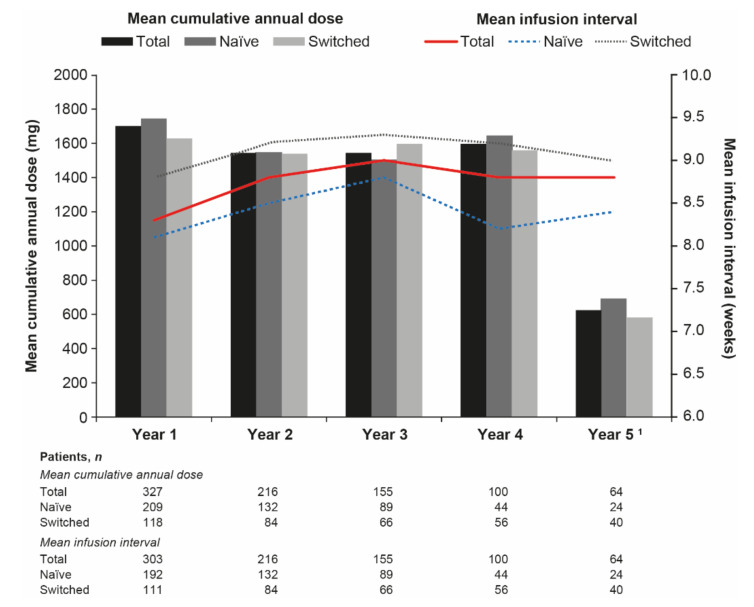
Mean cumulative annual maintenance dose and mean infusion interval by treatment group. Adapted with permission from Lee, S.S.; Kim, T.H.; Park, W.; Song, Y.W.; Suh, C.H.; Kim, S.; Yoo, D.H. (2021). © 2021 Authors (or their employer(s)) [21]. ^1^ Few patients were evaluable at Year 5.

**Figure 3 jcm-10-04568-f003:**
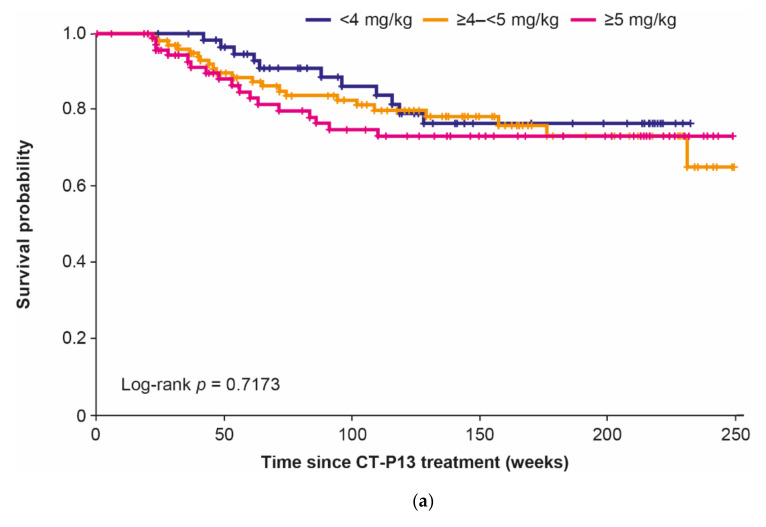
Drug survival by baseline dose group: (**a**) overall, (**b**) naïve patients, and (**c**) switched patients.

**Figure 4 jcm-10-04568-f004:**
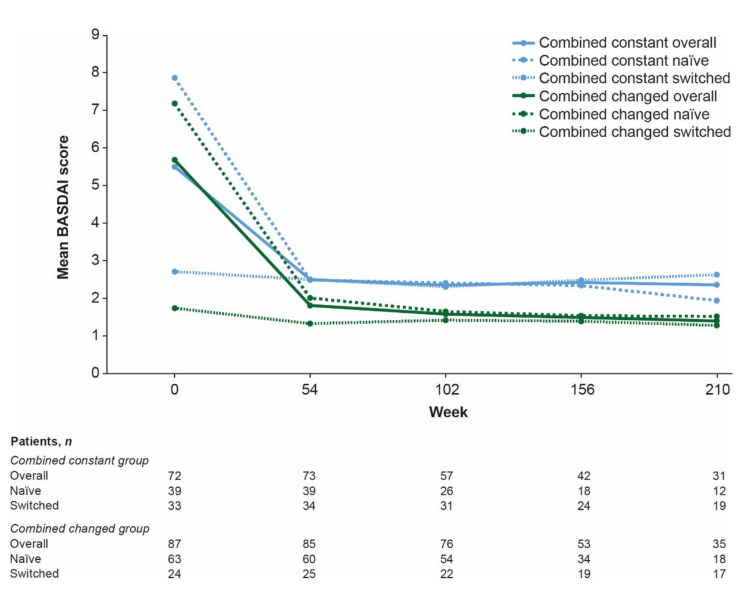
Mean BASDAI scores ^1^ by treatment group for combined constant and combined changed groups. ^1^ BASDAI results are not presented for week 264 as there was only one evaluable patient. BASDAI, Bath Ankylosing Spondylitis Disease Activity Index.

**Figure 5 jcm-10-04568-f005:**
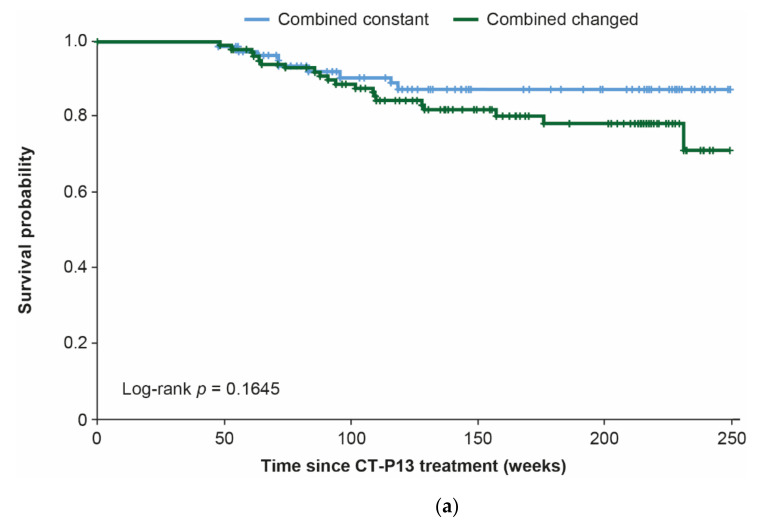
Drug survival for combined constant and combined changed groups: (**a**) overall, (**b**) naïve patients, and (**c**) switched patients.

**Table 1 jcm-10-04568-t001:** Baseline demographics and disease characteristics by baseline dose (overall population).

	Overall (*n* = 270 ^1^)	Baseline Dose			*p*-Value ^2^
		<4 mg/kg (*n* = 71)	≥4–<5 mg/kg (*n* = 117)	≥5 mg/kg (*n* = 82)	
Sex, *n* (%)					
Female	68 (25.2)	20 (28.2)	24 (20.5)	24 (29.3)	0.2987 ^3^
Male	202 (74.8)	51 (71.8)	93 (79.5)	58 (70.7)	
Median (IQR) treatment duration, months	24.2 (7.50–45.1)	24.3 (9.97–47.8)	23.8 (7.47–36.6)	24.2 (7.30–48.1)	0.6747
Median (IQR) disease duration, years	3 (1–6)	3 (1–6)	2 (1–7)	3 (1–6)	0.5956
Median (IQR) age, years	38 (30–51)	43 (31–54)	33 (27–44)	40 (32–51)	0.0013
Body weight, kg					
*n*	264	68	116	80	
Median (IQR)	67.0 (59.0–74.4)	66.1 (58.1–79.0)	68.0 (62.0–73.0)	60.0 (57.0–74.4)	0.0529
BMI, kg/m^2^					
*n*	223	63	85	75	
Median (IQR)	23.38 (21.10–25.71)	23.80 (22.44–26.93)	22.96 (21.10–24.68)	23.06 (20.69–25.90)	0.0633
BASDAI score					
*n*	212	55	88	69	
Median (IQR)	6.66 (3.86–7.90)	5.47 (0.58–7.20)	7.25 (4.80–8.15)	6.60 (4.73–8.00)	0.0007
ESR, mm/h					
*n*	259	69	113	77	
Median (IQR)	23 (7–50)	28 (11–58)	24 (8–47)	16 (5–40)	0.0882
CRP, mg/L					
*n*	259	69	113	77	
Median (IQR)	0.80 (0.19–2.64)	0.80 (0.27–2.72)	0.81 (0.31–3.40)	0.48 (0.08–1.60)	0.0436

^1^ Baseline dose was missing for 67 patients because of missing weight information (*n* = 57) or missing dose after induction period in naïve patients (*n* = 10). ^2^
*p*-values were determined by Kruskal–Wallis test, unless otherwise specified. ^3^
*p*-value was determined by chi-squared test. BASDAI, Bath Ankylosing Spondylitis Disease Activity Index; BMI, body mass index; CRP, C-reactive protein; ESR, erythrocyte sedimentation rate; IQR, interquartile range.

**Table 2 jcm-10-04568-t002:** Median (IQR) BASDAI scores ^1^ by treatment group for baseline dose analysis groups.

	Week 0	Week 54	Week 102	Week 156	Week 210
**<4 mg/kg baseline dose group**					
Overall					
*n*	55	48	39	28	22
Median (IQR)	5.47 (0.58–7.20)	0.88 (0.42–1.72)	0.80 (0.30–1.64)	0.49 (0.26–1.31)	0.60 (0.22–1.00)
Naïve					
*n*	30	24	17	12	8
Median (IQR)	7.00 (6.09–8.00)	1.15 (0.73–1.93)	1.30 (0.60–1.90)	1.31 (0.30–1.78)	0.97 (0.59–1.25)
Switched					
*n*	25	24	22	16	14
Median (IQR)	0.57 (0.30–1.66)	0.60 (0.30–1.10)	0.45 (0.30–1.20)	0.38 (0.26–0.62)	0.41 (0.18–0.97)
**≥4–<5 mg/kg baseline dose group**					
Overall					
*n*	88	68	59	40	21
Median (IQR)	7.25 (4.80–8.15)	2.00 (0.80–3.20)	1.57 (0.79–2.60)	1.40 (0.61–3.00)	1.60 (0.60–3.00)
Naïve					
*n*	67	49	41	26	11
Median (IQR)	7.50 (6.80–8.30)	2.00 (0.90–3.20)	1.30 (0.80–2.30)	1.25 (0.76–1.80)	1.60 (0.80–2.80)
Switched					
*n*	21	19	18	14	10
Median (IQR)	1.01 (0.66–2.80)	2.00 (0.49–3.40)	2.33 (0.50–3.20)	1.95 (0.43–3.20)	1.72 (0.36–3.40)
**≥5 mg/kg baseline dose group**					
Overall					
*n*	69	53	45	34	24
Median (IQR)	6.60 (4.73–8.00)	2.90 (1.40–4.40)	3.00 (1.20–4.00)	3.00 (2.00–4.20)	2.85 (1.60–4.00)
Naïve					
*n*	43	33	26	17	11
Median (IQR)	7.64 (6.50–9.10)	2.70 (1.30–4.20)	2.20 (0.94–3.80)	2.54 (0.80–3.60)	2.30 (0.80–3.60)
Switched					
*n*	26	20	19	17	13
Median (IQR)	4.24 (2.86–5.10)	3.30 (2.30–4.60)	3.10 (2.40–4.60)	3.10 (2.60–4.40)	3.00 (2.60–4.20)

^1^ BASDAI results are not presented for week 264 as there was only one evaluable patient. BASDAI, Bath Ankylosing Spondylitis Disease Activity Index; IQR, interquartile range.

## Data Availability

The data presented in this study are available in the current article and accompanying Appendix A.

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
