# Peer review of "Impact of Infliximab Biosimilar CT-P13 Dose and Infusion Interval on Real-World Drug Survival and Effectiveness in Patients with Ankylosing Spondylitis"

_jcm, 2021, doi:10.3390/jcm10194568_

Round 1

Reviewer 1 Report

1.Modfication of dose or interval is beneficial; but they were modified either by increasing or decreasing them; doesn't THE DIRECTION of modification have ANY INFLUENCE AT ALL on your results ? In other words IT DOES NOT MATTER if one would increase or decrease the dose or the interval ? The only thing that counts is JUST TO MODIFY any of the two ? I mean, for instance, if one would increase the dose, one would do so because of lack of efficacy; so how can this have the same better results as if one would decrease the dose ?

2.What was/were the reason/reasons for switching ? If there was no medical reason, then it was expected that switched patients would have a lower BASDAI than naive ones (since they were not switched for medical reasons). So, you must provide reason/reasons for switching. (rows 161-164 in your text). The same way of thinking applies to your results on rows 165-169: is was expected that switched patients would differ in their BADAI scores dependent on the baseline dose (those with a higher baseline BASDAI would be started on higher doses of infliximab, because they would be more active disease patients)

Reviewer 2 Report

No comments!

Very well written with excellent statistical analysis!

Author Response

Thank you for your feedback on our manuscript. We are pleased that you have found it valuable.